# Accurate Three-Dimensional Thermal Dosimetry and Assessment of Physiologic Response Are Essential for Optimizing Thermoradiotherapy

**DOI:** 10.3390/cancers14071701

**Published:** 2022-03-27

**Authors:** Mark W. Dewhirst, James R. Oleson, John Kirkpatrick, Timothy W. Secomb

**Affiliations:** 1Department of Radiation Oncology, Duke University School of Medicine, Durham, NC 27710, USA; jncoleson@bellsouth.net (J.R.O.); john.kirkpatrick@duke.edu (J.K.); 2Department of Physiology, University of Arizona, Tucson, AZ 85724, USA; secomb@u.arizona.edu

**Keywords:** thermal dosimetry, hypoxia, hyperthermia, radiation therapy, reoxygenation, perfusion, oxygen consumption rate, local tumor control, biomarker

## Abstract

**Simple Summary:**

Many clinical trials have shown benefit for adding hyperthermia (heat) treatment to radiotherapy. Despite overall success, some patients do not derive maximum benefit from this combination treatment. Tumor hypoxia (low oxygen concentration) is a major cause for radiotherapy treatment resistance. In this paper, we examine the question of whether hyperthermia reduces hypoxia and, if so, whether reduction in hypoxia is associated with treatment outcome. The review is focused mainly on several clinical trials conducted in humans and companion dogs with cancer treated with hyperthermia and radiotherapy. Detailed measurements of temperature, hypoxia and perfusion were made and compared with treatment outcome. These analyses show that reoxygenation after hyperthermia occurs in patients and is related to treatment outcome. Further, reoxygenation is most likely caused by variable intra-tumoral temperatures that improve perfusion and reduce oxygen consumption rate. Directions for future research on this important issue are indicated.

**Abstract:**

Numerous randomized trials have revealed that hyperthermia (HT) + radiotherapy or chemotherapy improves local tumor control, progression free and overall survival vs. radiotherapy or chemotherapy alone. Despite these successes, however, some individuals fail combination therapy; not every patient will obtain maximal benefit from HT. There are many potential reasons for failure. In this paper, we focus on how HT influences tumor hypoxia, since hypoxia negatively influences radiotherapy and chemotherapy response as well as immune surveillance. Pre-clinically, it is well established that reoxygenation of tumors in response to HT is related to the time and temperature of exposure. In most pre-clinical studies, reoxygenation occurs only during or shortly after a HT treatment. If this were the case clinically, then it would be challenging to take advantage of HT induced reoxygenation. An important question, therefore, is whether HT induced reoxygenation occurs in the clinic that is of radiobiological significance. In this review, we will discuss the influence of thermal history on reoxygenation in both human and canine cancers treated with thermoradiotherapy. Results of several clinical series show that reoxygenation is observed and persists for 24–48 h after HT. Further, reoxygenation is associated with treatment outcome in thermoradiotherapy trials as assessed by: (1) a doubling of pathologic complete response (pCR) in human soft tissue sarcomas, (2) a 14 mmHg increase in pO2 of locally advanced breast cancers achieving a clinical response vs. a 9 mmHg decrease in pO2 of locally advanced breast cancers that did not respond and (3) a significant correlation between extent of reoxygenation (as assessed by pO2 probes and hypoxia marker drug immunohistochemistry) and duration of local tumor control in canine soft tissue sarcomas. The persistence of reoxygenation out to 24–48 h post HT is distinctly different from most reported rodent studies. In these clinical series, comparison of thermal data with physiologic response shows that within the same tumor, temperatures at the higher end of the temperature distribution likely kill cells, resulting in reduced oxygen consumption rate, while lower temperatures in the same tumor improve perfusion. However, reoxygenation does not occur in all subjects, leading to significant uncertainty about the thermal–physiologic relationship. This uncertainty stems from limited knowledge about the spatiotemporal characteristics of temperature and physiologic response. We conclude with recommendations for future research with emphasis on retrieving co-registered thermal and physiologic data before and after HT in order to begin to unravel complex thermophysiologic interactions that appear to occur with thermoradiotherapy.

## 1. Introduction

Key meta-analyses have been published on locally advanced cervix cancer [1], head and neck cancer [2] and chest wall recurrences of breast cancer [3], showing therapeutic benefit in terms of improvement of either/or local tumor control, progression free and overall survival after combining local-regional HT with radiotherapy. An important randomized trial comparing multi-agent chemotherapy +/− HT showed improvements in progression free and overall survival in patients with locally advanced high-risk soft tissue sarcomas in the arm receiving HT [4,5].

Despite the overall success of many trials, a therapeutic benefit was not obtained in all patients and some randomized trials did not show a statistically significant therapeutic benefit of HT [6,7,8]. Even in those patients in which there was some benefit, it may not have been maximally optimized. Demonstration of enhanced anti-tumor effect with HT would increase its wider acceptance as a viable adjuvant therapy. Thus, there is strong rationale for investigating mitigating factors that may play a role in treatment outcome.

HT induces a number of biologic and physiologic effects on tumors. HT inhibits multiple DNA damage repair mechanisms, which play a major role in heat radiosensitization. The inhibition of DNA repair provides a rationale for combining HT with HSP90 (heat shock protein-90) and/or PARP (poly (ADP-ribose) polymerase) inhibitors [9]. Heat shock proteins, HSP70 and HSP27, bind to enzymes to facilitate base excision repair [10]. This heat shock protein association may enhance DNA damage repair after HT. Substantiating this hypothesis is the observation that enhancement of repair of heat induced double strand breaks is linked to HSP70 and HSP27 association with heat labile DNA polymerase beta in thermotolerant cells [11]. The thermotolerance-induced enhancement of DNA damage repair could reduce the effectiveness of radiotherapy treatments administered when cells are thermotolerant [12,13]. If so, such an effect could reduce the impact of reoxygenation observed 24–48 h post HT, which is the main subject of this review. It is unknown whether this mechanism of thermotolerance-induced radioresistance is clinically relevant. Further research would be needed to answer this question.

Maximal thermal enhancement of radiotherapy in pre-clinical and theoretical models occurs when the two modalities are given simultaneously or within a short time interval between the two [14]. The effect of time interval on radiosensitization is the result of the effects of HT on DNA damage repair [14]. Retrospective analysis of the impact of time interval between HT and radiotherapy has been controversial for cervix cancer [15,16,17,18]. A call for standardization of methods and results reporting has been recently published [19]. Standardization of reporting will contribute significantly toward understanding how to optimize thermoradiotherapy from the perspective of methods of delivery and documentation of results.

Hyperthermia also induces a number of immunostimulatory effects in both the innate and adaptive immune systems [20] that are likely important for its biological effectiveness when combined with radiotherapy. HT is cytotoxic itself, with the extent of cytotoxicity being dependent upon the time and temperature of heating [21]. Further, the cytotoxicity of HT is not dependent upon oxygen availability, so it is complementary to radiation in this respect, since hypoxia causes significant reduction in cytotoxicity of radiotherapy [22].

In this review, we will focus on the clinical observation that HT can reduce hypoxia up to at least 1–2 days after HT. Further, the reoxygenation is associated with treatment outcome in patients with locally advanced breast cancer and soft tissue sarcomas in humans and in companion dogs. These observations suggest that positive interactions between HT and radiotherapy can occur outside the short time window suggested for maximal interaction from pre-clinical studies.

Tumor hypoxia is well-established as a cause for radioresistance and treatment failure [23,24,25,26]. Hypoxia is also known to negatively influence treatment response to chemotherapy [27] and immunotherapy [20,28], as well as contributing to tumor aggressiveness [29,30,31,32,33]. A recent *Special Issue in Cancers* contained several original reports and contemporary review papers on the subject of tumor hypoxia [34,35,36,37,38,39,40,41,42,43,44,45,46,47,48]. In this review, we will consider how thermal dose affects tumor hypoxia and, in turn, whether changes in hypoxia in response to thermoradiotherapy can influence treatment outcome.

Extensive pre-clinical studies have been conducted in tumor-bearing rodents with cancer, and these studies revealed important trends in defining the relationship between conditions of thermal exposure and changes in perfusion and hypoxia [49,50,51]. It has been shown that heating rates in the range of 1 °C/min are: (1) more cytotoxic in vitro [52] and (2) more damaging to tumor microvasculature than slower heating rates [53]. Further, reduced perfusion and enhanced anti-tumor effect after HT alone has been shown to be associated with faster heating rates [54]. It is unknown whether faster heating rates impact reoxygenation 24–48 h post HT in either pre-clinical models or clinically. Heating rate effects have not been studied in conjunction with radiotherapy. If faster heating rates cause vascular damage and hypoxia, then they may result in radioresistance.

For the most part, pre-clinical studies were not designed to test whether changes in perfusion and hypoxia in individual subjects were associated with individual treatment outcome. Such information is required for perfusion or hypoxia measurements to be clinically translatable. Therefore, we will review studies conducted primarily in humans and companion dogs with cancer, where detailed thermometry and physiologic data were acquired for each individual. In most cases, treatment outcome was also documented.

For the purposes of this review, we define 30–60 min of “mild heating” as temperatures from 40 to 42 °C, because minimal direct cell killing occurs in this range. A number of other effects occur in this temperature range, however, including increases in perfusion [22,55] and vascular permeability [56], alterations in cell signaling [9,57,58,59], inhibition of DNA damage repair [9], inhibition of the HPV viral oncoprotein, E6 [60] and immunologic effects [20]. “Moderate heating” is defined as temperatures >42 and <44 °C. In this moderate temperature range, direct thermal cytotoxicity occurs [61], in addition to many of the effects described above in the mild heating range. “High heating” occurs at temperatures >44 °C and <50 °C. We truncate the high temperature heating at 50 °C to distinguish it from thermal ablation, which occurs at temperatures higher than 60 °C. We have adopted this classification because temperatures >44 °C can increase tumor hypoxia in canine soft tissue sarcomas, whereas below this threshold, hypoxia is either not affected or is reduced [62,63]. Others have used adjectival descriptors of mild (40–42 °C), moderate (42–45 °C) and T > 45 °C as causing irreversible damage [64]. This classification is similar to what we describe. We have chosen 30–60 min heating because that is the range over which HT is most often administered clinically.

## 2. Hypoxia Is Caused by Imbalance between Oxygen Delivery and Oxygen Consumption Rate

The pO2 of any location within a tissue is governed by the balance between oxygen delivery and oxygen consumption. Oxygen delivery is influenced by the flow rate of microvessels, oxygen content, vascular density and vessel orientation surrounding the location [65]. An important question to ask is which of these factors has the greatest influence on development of hypoxia. Computer generated sensitivity studies were used to address the question of whether increasing oxygen delivery or reducing oxygen consumption rate would be more effective in reducing tumor hypoxia [66,67]. These simulations were based on in vivo measurements of the parameters listed above. Reducing oxygen consumption rate was more efficient by factors of 10–30-fold, compared with increasing blood flow rate or oxygen content of blood, respectively [66]. It has been shown in vitro that elevation of glucose concentration reduces oxygen consumption rate as cells switch to anaerobic metabolism. Induction of hyperglycemia with hyperoxic gas breathing was synergistic in reducing tumor hypoxia in computer simulations [68] and in vivo [69]. Similarly, the combination of HT and carbogen breathing was shown to significantly increase tumor pO2 and enhance radiotherapeutic response [70,71]. HT can also affect oxygen consumption rates, so it is important to consider such effects when evaluating how HT affects tumor hypoxia.

In this review, we address questions about effects of HT on:hypoxia,perfusion,metabolism and oxygen consumption rate andnecrosis.

Some pre-clinical data will be presented as background. However, the main focus will be on what clinical evidence exists for HT affecting factors that influence tumor hypoxia and whether such changes influence thermoradiotherapeutic treatment outcome.

## 3. Challenges to Relating Temperatures Achieved during HT with Physiologic Response

### 3.1. Difference in Temperature Distributions between Rodent and Human Tumors

In rodent tumors, water bath heating exposes the skin and normal tissue around the tumor to the highest temperatures because they are immediately adjacent to the water in the bath; intra-tumoral temperatures are somewhat lower and relatively uniform [72]. In human tumors, there can be large variations in temperature (several degrees above and below the median value) within tumors. The tumor margin and surrounding normal tissue may not be heated appreciably, while the interior of the tumor is hotter [73]. The spatial variation in temperature in human tumors is related to non-uniformities in power deposition from heating devices, with spatial variations in: (1) tissue properties and (2) peri- and intra-tumoral perfusion [74,75,76,77,78]. The differences in the temperature distribution between rodent and human tumors may contribute to differences in physiologic response to HT (Figure 1).

### 3.2. Thermometry in Human Tumors Is Mainly Acquired from Implanted Thermal Probes

Since temperatures in human tumors are heterogeneous, thermometry is essential to assess the therapeutic value of a treatment. The vast majority of clinical thermal data to date has been derived from direct intra-tumoral measurements. Typically, one to two catheters are placed into the tumor and temperatures are measured as thermometers are pushed back and forth within the catheter [82]. The resultant data are depicted by descriptors of the temperature distribution, such as T_90_ [10th percentile of distribution], T_50_ [distribution median] or T_10_ [90th percentile] [82]. Descriptors of the temperature distribution do not reveal anything about the spatial distribution of temperature, but rather provide an overall summary for the tumor as a whole. Non-invasive thermometry can provide spatially encoded thermal data, and this method has been implemented in some patients [78,81,82]. In the future, combinations of non-invasive thermometry with imaging of physiologic response may reveal whether intra-tumoral heterogeneity of physiologic response in tumors is dictated by local temperature variation.

## 4. Effects of Hyperthermia on Tumor Metabolism

It has been reported previously that enzyme activity increases with temperature and time of heating until the point where enzyme denaturation occurs [83]. These effects are observed during heating and could influence oxygenation during HT. However, effects occurring during HT may not be related to what happens 24–48 h later. There are two documented effects in tumors after HT that could influence oxygen consumption rate: (1) switch to anaerobic metabolism and (2) direct cytotoxicity by hyperthermia.

### 4.1. Switch to Anaerobic Metabolism after Hyperthermia Treatment

Kelleher utilized a near-IR heating device to heat DS-sarcomas in rats for 60 min [84]. This device yielded temperature distributions analogous to what is seen clinically, with T_90_, T_50_ and T_10_ values of 42.6, 43.8 and 44.8 °C, respectively. Using a bioluminescence method in snap frozen tissues, lactate and glucose levels were significantly increased, whereas ATP concentrations were decreased after HT. The depletion in ATP concentration is consistent with a reduction in oxidative phosphorylation, whereas the increase in lactate concentration is consistent with a switch to anaerobic metabolism. This switch to anaerobic metabolism is associated with reduction in oxygen consumption rate.

Others have used 31-P Magnetic resonance spectroscopy to monitor ATP concentrations immediately after HT at various temperatures and times of heating [85,86]. They showed significant temperature and heating time-dependent reductions in ATP/Pi (Pi = inorganic phosphate) ratio at temperatures between 43 and 44 °C. In canine sarcomas, depletion in ATP/Pi ratio at 24 h post HT was dependent upon CEM43T_50_ and CEM43T_90_ during heating [87]. Further, reduction in ATP/PME [phosphomonoester] was significantly correlated with probability of pathologic complete response rate (pCR rate) in humans with soft tissue sarcomas [87]. Although the time intervals after HT when measurements were made in rodents and these spontaneous sarcomas are different, there is remarkable similarity in the temperature dependence of ATP depletion.

We conducted a phase II study in human soft tissue sarcomas, where we hypothesized that reaching a pre-determined thermal dose would lead to >75% incidence of pCR rate [88]. We failed to prove the hypothesis, but in parallel studies conducted in the same patient series, we found that pre-treatment metabolic factors, such as hypoxia, phosphodiester/inorganic phosphate (PDE/Pi) and phosphomonoester/Pi (PME/Pi) ratios, were associated with pCR rate [89]. We speculated that in this particular trial, pre-treatment physiology interfered with our ability to show the hypothesized thermal dose–response relationship.

Moon et al. examined potential underlying mechanisms for the apparent switch to anaerobic metabolism after 42 °C HT [57]. HT increased hypoxia inducible factor-1α (HIF-1α) for several hours after HT. HIF-1 is a heterodimer, consisting of HIF-1α and HIF-1β subunits. When bound together, HIF-1 enters the nucleus and initiates transcription of many genes, including PDK1 (3-phosphoinositide-dependent kinase 1), which controls the switch to anaerobic metabolism. Normally, HIF-1α is efficiently degraded by prolyl hydroxylases that initiate degradation of HIF-1α so that the heterodimer does not form [65]. HIF-1α is stabilized during hypoxia because the prolyl hydroxylases require oxygen for their action. However, in the case of HT, inactivation of HIF-1α degradation was associated with an increase in oxidative stress. The switch to anaerobic metabolism would reduce oxygen consumption rate, since anaerobic metabolism does not rely on oxygen to produce ATP.

Radiotherapy is also known to increase HIF-1 dependent transcription, but underlying mechanisms for HIF-1 upregulation are different from HT and are radiation dose dependent. For doses in the range of conventionally fractionated radiotherapy, HIF-1 dependent transcription is upregulated in response to increased oxidative stress associated with reoxygenation [90], followed by prolonged HIF-1 upregulation in response to massive nitric oxide production by infiltrating macrophages [91]. Higher single radiotherapy doses, in the range of 15Gy, decrease perfusion and increase hypoxia by causing microvascular damage; HIF-1 dependent transcription is subsequently upregulated by hypoxia [92]. Mild temperature heating immediately after high dose radiation reduces the radiation induced upregulation of HIF-1α caused by vascular damage by radiotherapy [92]. These differing effects of HT and radiotherapy dose on HIF-1 expression may be important in affecting tumor metabolism and treatment response.

Another method for assessing metabolic response to HT is 18-FDG-PET. Glucose uptake would be expected to increase if there is a switch to anaerobic metabolism, in the absence of extensive tumor cell killing by treatment. Some studies have been conducted in human patients prior to and after HT. However, these reports involved repeat scans taken weeks into the treatment course or even after treatment was completed. These studies showed that reductions in 18-FDG-PET uptake are associated with pathologic response in patients with esophageal cancer [93], rectal cancer [94] and soft tissue sarcomas [95]. The results are more likely dominated by extent of cell killing than by HT induced changes in cellular glucose uptake.

### 4.2. Direct Cytotoxicity of HT

The cytotoxic effects of HT are logarithmically related to temperature and linearly to the time of heating [96]. Sapareto and Dewey were the first to develop means to relate any time–temperature history into an equivalent number of minutes of heating at 43 °C [61]. This formulation has proven useful in describing tissue damage across a range of tissue types and temperature time histories as long as temperature is less than 50 °C [21,96]. The acronym for cumulative equivalent number of minutes at 43 °C is referred to as CEM43. An important question is whether there is enough direct cytotoxicity from HT to influence oxygen consumption rates.

Rosner et al. [97] conducted a theoretical study asking how much cell killing would be expected from a non-uniform temperature distribution typical of what is observed clinically. The temperature distributions were derived from a finite element heat transfer model of a simulated subcutaneous tumor, where power was delivered from a microwave applicator. Cytotoxicity was predicted based on a stochastic model of cell killing probability, based on survival curve data from CHO cells. For 60 min HT, the simulations revealed that 30–50% of cells would be directly killed by HT with a T_90_ of 41 °C. This occurs because of cell killing temperatures higher than the T_90_. Simulated temperatures above the T_90_ ranged up to 45.5 °C. Thermal killing of 30–50% of tumor cells would be sufficient to have an important impact on oxygen consumption rate and tumor hypoxia [66].

Below, we provide additional clinical results, addressing the question of whether increases in perfusion and/or direct cell killing by HT contributes to reoxygenation.

## 5. Effects of Hyperthermia on Tumor Perfusion and Hypoxia

Most of the published pre-clinical data have focused on effects of HT on perfusion and hypoxia during or immediately after treatment. However, there is a second body of work that has focused on effects that occur 24–48 h after treatment. Both will be discussed.

### 5.1. Physiologic Effects during or Immediately after Heating

The effects of HT on tumor perfusion and hypoxia have been studied extensively at the pre-clinical level. Pre-clinical data demonstrate an increase in perfusion and oxygenation during and shortly after heating at mild temperatures (39–42 °C) at heating times of 30–60 min [98,99]. At temperatures >43–46 °C for 30–60 min there is significant damage to vasculature, leading to hypoxia, anoxia and necrosis [100]. Thus, at the pre-clinical level, the physiologic response of tumors during or immediately after HT is bi-phasic. If reoxygenation occurs only during the application of HT, then taking advantage of it with radiotherapy would require simultaneous application of radiotherapy with HT. 

### 5.2. Physiologic Effects Occurring after Heating

In his Robinson Award manuscript, Oleson hypothesized that the enhanced effectiveness of HT + radiotherapy compared with radiotherapy alone had to be a result of reoxygenation [101]. The effectiveness of radiotherapy fractions given 24 h after HT could be influenced by HT induced reoxygenation. Part of his rationale was based on the observation that the prognostically important temperatures from clinical trials are at the lower end of the temperature distribution, where little direct cell killing occurs. Subsequent to Oleson’s paper, several papers were published, showing results that are consistent with his hypothesis.

Shakil et al. [98] were the first to report on reoxygenation occurring 24 h after mild temperature water bath HT of the R3230Ac rat mammary tumor to 40.5–43.5 °C for 30–60 min. Perfusion increased by 10–33% at the end of 30 min HT. At 24 h post HT, perfusion was further increased by two-fold over baseline. Immediately after HT, pO2 values increased two-fold, compared with baseline. At 24 h post HT, pO2 remained elevated, although lower than that seen immediately after HT. Similar effects were seen in other tumor models [99,102].

It has been speculated that reoxygenation rarely occurs hours to days after HT in human subjects; if it does occur, it has little to do with enhancing cell killing by radiotherapy [14]. Given the complexity of physiologic effects that occur in tumors in response to HT, this challenge requires rigorous and critical thought. This question will be addressed in the following discussion of clinical results.

### 5.3. Human Studies of Reoxygenation Post HT

Brizel et al. [103] reported that reoxygenation occurs at 24 h post heating in a portion of 38 patients with soft tissue sarcomas treated with pre-operative thermoradiotherapy (50 Gy in 2 Gy fractions, 5 fractions per week and 1–2 fractions of HT per week, given 1–2 h post radiotherapy). Oxygenation (Eppendorf pO2 histography) did not change after the first week of conventionally fractionated radiotherapy. However, median pO2 24–48 h after the first HT (given during second week of radiotherapy) increased from 6.2 mmHg to 12.4 mmHg, which was statistically significant. There was a significant correlation between reoxygenation and percent necrosis in the resected tumors. The median T_90_ in these tumors was 39.9 °C in tumors that had <90% necrosis, vs. 40.0 °C for tumors that achieved >90% necrosis [pathologic complete response; pCR—this small difference was not significant]. T_90_ values were lower than temperatures required for direct cell killing by HT [96]. This argues against the idea that pCR was a result of direct cell killing by HT, as hypothesized by others [14]. Although the results are provocative, a rigorous examination between thermal dose achieved and extent of reoxygenation and treatment outcome was not undertaken in this series.

Vujaskovic reported on a series of women with locally advanced breast cancer who received neoadjuvant chemotherapy consisting of liposomal doxorubicin [Myocet^TM^ and paclitaxel] combined with HT [104]. The rationale for this treatment was to take advantage of effects of HT on vascular permeability and liposomal extravasation [105,106]. pO2 measurements were made, using Eppendorf pO2 histography, prior to and 24 h after the second HT, which coincided with the second chemotherapy treatment course. Eleven of eighteen tumors were hypoxic (median pO2 < 10 mmHg). In the hypoxic tumors, eight out of eleven exhibited reoxygenation [median pO2 = 19.2 mmHg]. The response rate for hypoxic tumors that reoxygenated was higher than a sub-group that did not reoxygenate. There was no correlation between extent of reoxygenation and thermal dose in this group of patients, but there was a trend indicating that chances of reoxygenation were greater if median T_50_ remained between 39.5 and 41 °C [104]. This trend, showing a better chance for response with relatively low T_50_ values, was consistent with a separate group of patients with locally advanced breast cancer who were treated with pre-operative HT, radiotherapy and taxol [107]. Tumors that achieved either a partial or complete response were well oxygenated at baseline or reoxygenated by a median of 18 mmHg. Those tumors that had no response to treatment showed a reduction of pO2, by a median of 9 mmHg. In this clinical series, temperatures were not high enough to cause appreciable direct cell killing by HT.

### 5.4. Canine Studies of Reoxygenation Post HT

Vujaskovic also reported on changes in tumor oxygenation in a series of 13 dogs with soft tissue sarcomas treated with thermoradiotherapy [62]. Oxygen measurements were made prior to and 24 h after the first HT. The Oxford Optronix™ fluorescence lifetime probe was used to measure pO2 in multiple locations by placing the probe deep in the tumor and then recording pO2 during using a pull-back. Reduction in hypoxic fraction (HF) was observed for T_50_ values ranging from 39.5 to 44 °C. HF increased when T_50_ values were >44 °C. Consistent with the human studies, mild temperature HT improved tumor oxygenation, whereas higher temperatures contributed to apparent vascular damage, with an increase in tumor hypoxia. In this study, correlations between the oxygenation measurements with treatment outcome were not made.

Thrall et al. [108] conducted a randomized thermal dose escalation clinical trial that compared long term local tumor control in 122 dogs with soft tissue sarcomas that were randomized into two different thermal dose groups in combination with fractionated radiotherapy (2.25 Gy/fx, 25Fx). There was a 17-fold higher CEM43T_90_ in the high vs. the low HT dose group (Figure 2A). The difference in thermal dose was achieved by generating higher temperatures and longer heating times in the high thermal dose group (Figure 2B,C). Duration of local tumor control was significantly longer in the high thermal dose group, with a hazard ratio of 2.3 in multivariate analysis.

Hypoxia was measured in subgroups of animals in this trial. These results have not been published previously. The Oxford Optronix™ fluorescence lifetime probe was used prior to and 24 h after the first HT to determine change in median pO2 and HF in 11 subjects (% measurements < 10 mmHg). There were significant correlations (Pearson correlation) between increased median pO2 (*p* = 0.0230) or reduced HF (*p* = 0.007) and duration of local control (Table 1). This observation was corroborated in another subgroup of 16 animals that were given pimonidazole prior to and 24 h after the first HT. Immunohistochemistry was used to determine the hypoxic fraction, as described by Cline et al. [110,111]. Reduction in the % pimonidazole positive area was inversely associated with increased time to local failure. Caution has to be used, given the small number of patients in these analyses. However, the similarity between the oxygen probe results and the pimonidazole data suggest that reoxygenation after the first HT is likely predictive of time to local failure. Additional studies would be required for validation.

Thrall et al. reported on another trial of 37 dogs with soft tissue sarcomas that were treated with two different HT dose fractionation schedules (5Fx (*n* = 21) vs. 20Fx (*n* = 16)), in conjunction with fractionated radiotherapy (2.25 Gy/Fx, 25Fx) [63]. The goal of this thermal dose fractionation trial was to achieve equivalent CEM43T_90_ for both fractionation schedules. The working hypothesis was that the 20Fx group would achieve better anti-tumor effect compared with the 5Fx group. In the final analysis, CEM43T_90_ was slightly and significantly higher in the 5Fx HT arm (29.9 vs. 24.9 CEM43T_90_ for the 5 vs. 20 HT fractions, respectively). To accomplish near equivalence in total CEM43T_90_ between the treatment groups, the duration of heating for the 5Fx HT group was six-fold longer per treatment. Although T_50_ and T_10_ values were higher for the 5Fx HT group than the 20Fx HT group, the total CEM 43 T_10_ and T_50_ values were higher in the 20Fx HT group. This was a product of the larger number of HT fractions in this group (Appendix A). Multiple physiologic endpoints were measured in these subjects, pre and 24 h after the first HT: pO2, contrast enhanced perfusion with MRI, apparent diffusion coefficient (ADC) with MRI, and genomic analysis [112]. Contrary to the hypothesis, the 5Fx HT group showed greater volume reduction than the 20Fx HT group (*p* = 0.0022). The physiologic endpoints associated with treatment group were change in ADC after the treatment course and change in perfusion at 24 h after the first HT. Additionally, there was a significant correlation between HF change 24 h after the first HT and tumor volume change at the end of therapy; as hypoxic fraction was reduced, tumor volume was reduced. The 5Fx HT group showed a trend toward a reduction in ADC. In contrast, the 20Fx HT group showed increased ADC values (Appendix A). Increases in ADC values at the end of therapy were associated with changes in gene expression at 24 h post first HT, consistent with induction of inflammation [112]. Thus, the increase in ADC with the 20Fx HT group may be associated with increased edema as a result of inflammation. There was also a significant difference in perfusion response after the first HT between the two arms. The 5Fx HT arm exhibited increases in perfusion, whereas the 20Fx HT arm exhibited decreases in perfusion.

Further analyses of data from this trial, which have not been published previously, suggest that the reoxygenation observed in these tumors is linked to the distribution of thermal dose. The results of this analysis are shown in Table 2 and Table 3.
Higher CEM43T_10_ was associated with an improvement in average pO2 (*p* = 0.0214) and reduction in HF (% points < 10 mmHg; *p* = 0.0451), 24 h after the first HT.There was a significant positive correlation between CEM43T_90_ and perfusion at 24 h post first hyperthermia fraction.Increases in average pO2 and perfusion at 24 h after the first HT were correlated with tumor volume reduction at the end of treatment.Higher Total CEM43T_10_ and Total CEM43T_50_ were associated with change in ADC at the end of treatment (*p* = 0.007 and *p* = 0.0007, respectively), but the trends were different for the 5Fx HT vs. 20Fx HT groups. Reduction in ADC is associated with lower diffusion coefficient of water, which can be interpreted as a relative decrease in water mobility. It has been reported that early onset of apoptosis or apoptosis mixed with necrosis is associated with increased ADC [113,114]. However, in situations where there is necrosis in the absence of apoptosis, chronic necrosis or fibrosis, ADC tends to decrease [115,116]. The increase in ADC associated with relatively high CEM43T_10_ and -T_50_ in the 20Fx HT group is consistent with the notion that higher cumulative thermal doses cause cell killing and increased edema. Extensive cell death could reduce oxygen consumption rate across a tumor, thereby contributing to improved oxygenation.Higher CEM43T_10_ and -T_50_ were significantly negatively correlated with greater tumor volume reduction at the end of therapy.

These results provide a direct link between characteristics of the temperature distribution, potential mechanisms of reoxygenation and treatment response. We hypothesize that the reduction in hypoxia is associated with a reduction in oxygen consumption rate associated with the higher end of the thermal dose distribution (CEM43T_10_, -T_50_), combined with an increase in perfusion associated with the lower end of the temperature distribution (CEM43T_90_) (Figure 3).

There are some conundrums in the results, however. Contrary to the correlation between T_10_ and ADC change at the end of treatment, there was no correlation between T_10_ and ADC change at 24 h post treatment [117]. These results could be interpreted as indicating that cell killing does not contribute to reoxygenation 24 h after the first HT. It is possible that this lack of correlation of T_10_ with ADC change at 24 h post HT has to do with the relatively small volume of tumor represented by the T_10_ (90% of measurements would be <T_10_). If there was direct cytotoxicity after the first HT in the volume represented by the T_10_, it may not have impacted the overall median ADC. Another option to consider is that temperatures >T_50_ interfered with respiration, thereby reducing oxygen consumption rate. As discussed earlier in this review, respiration is relatively thermosensitive and is reduced in the temperature range of T_10_ and T_50_ (see Figure 2). A reduction in oxygen consumption rate, even in a small sub-volume of the tumor, would be sufficient to impact oxygen transport and reduce hypoxic fraction. Additional evidence for a reduction in oxygen consumption rate as a contributor to reoxygenation comes from observations that HIF-1 regulated genes and proteins were upregulated after HT [63,112] in these subjects. Increases in HIF-1 would cause a switch to anerobic metabolism [57,118]. Finally, it was not possible to follow these individuals to ascertain long term local tumor control or progression free survival. Clearly, further research is required.

**Figure 3 cancers-14-01701-f003:**
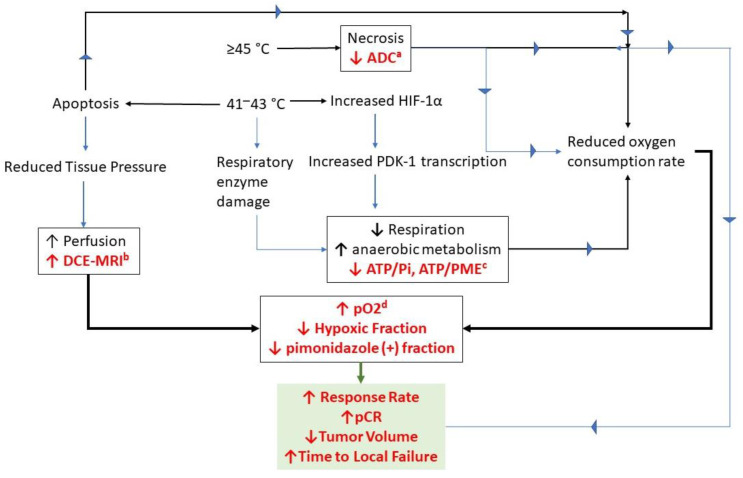
Potential mechanisms for reoxygenation following HT. The boxes in this figure contain putative mechanisms for reoxygenation, along with supportive data acquired from canine soft tissue sarcomas and/or human patients with soft tissue sarcomas or locally advanced breast cancer. The terms highlighted in red font are observations that support the proposed mechanisms. The box highlighted in green lists treatment responses that are linked back to the physiologic response observations. The superscripted letters next to the individual measurements refer back to the papers in which the observations were reported. a—[63]; b—[62,63]; c—[87]; d—[62,63,104,107,119]. The temperatures listed are linked to typical heating times of 60 min per HT fraction.

Viglianti et al. [120] examined tumor perfusion using DCE/MRI [dynamic contrast enhanced MRI] prior to and 24 h post first HT in the canine soft tissue sarcomas treated with thermoradiotherapy. Perfusion was measured prior to and 24 h after the first HT, [120]. Although perfusion increased in some subjects after HT, there was no association with local tumor control. Vaupel suggested that integrated temperature–time combination could be associated with biphasic vascular effects of HT [51]. Further work would be needed to verify that physiologic effects are associated with this measure of thermal dose. The integrated time–temperature approach has been reported to be associated with treatment outcome to thermoradiotherapy, however [121,122].

Recently, Thomsen et al. [123] reported on changes in oxygenation of the chest wall skin of normal subjects and patients with chest-wall recurrences of breast cancer. Water-filtered infrared-A-irradiation was used to heat this superficial tumor site. Hyperspectral imaging was used to ascertain hemoglobin saturation. Implanted fiber optic oxygen sensors (Oxford Optronix™, fluorescence life time probe) were used to measure pO2 directly. In normal volunteers, tissue oxygenation increased during HT to reach an elevated plateau and slowly declined after power was turned off. Measurements of Hb_sat_ followed a similar pattern, with elevations persisting up to 15 min post heating [123]. Preliminary patient data were also provided, suggesting a similar time course for change in oxygenation. These data are provocative. We await follow up reports as to whether improvements in oxygenation in these tumor bearing subjects are associated with treatment outcome.

Waterman et al. [124] measured perfusion in superficial human tumors during HT using a thermal diffusion method based on monitoring the rate of decline in temperature during brief periods of turning off microwave applicator power. He also observed increases in perfusion during heating [124]. These patients were treated with thermoradiotherapy, but the authors did not report whether the changes in perfusion were associated with tumor response. 

Thrall et al. [119] reported on changes in tumor hypoxia in a series of seven dogs over a five-week course of thermoradiotherapy. Hypoxia was measured using the Oxford Optronix™ fluorescence lifetime probe 3–4× per week. In four out of five tumors that were hypoxic at baseline, reduction in hypoxia observed after the first HT continued to be observed throughout the treatment course. This included measurements that were made during several day intervals when HT was not administered. In a fifth marginally hypoxic tumor at baseline, pO2 values dropped to near zero at 24 h post first HT and remained that way for the duration of the treatment course. The remaining three tumors were not hypoxic to start with and the treatment course did not cause hypoxia. In this series of tumors, T_90_ values were far below those that would cause appreciable direct cell killing by HT. 

## 6. A Look Backward and Future Directions

As indicated in the beginning of this review, there were concerns raised as to whether reoxygenation occurs in 1–2 days after HT and, if so, whether it has any influence on radiobiologically significant hypoxia [14]. We can say without reservation that reoxygenation can occur up to 24 h and perhaps even longer after HT. We showed this was the case in: (1) human soft tissue sarcomas [103], (2) four separate series involving canine soft tissue sarcomas [62,63,109,119] and (3) in two clinical trials of women with locally advanced breast cancer [104,107]. Concerns were raised as to whether clinical responses, such as pathologic CR rate, were simply caused by HT induced necrosis as opposed to reoxygenation having an effect on radiosensitivity [14]. Although we show clear evidence that CEM43T_10_ and CEM43T_50_ are associated with necrosis induction, temperatures at the lower end of the distribution are too low to cause direct cell killing by HT (Figure 2 and Appendix A). Similar results were reported previously in human sarcomas [73]. Thus, it seems implausible to explain complete pathologic response or early tumor response by simple necrotic cell killing, as has been suggested by others [14].

We have speculated that reoxygenation occurs as a result of direct HT cytotoxicity of aerobic cells, which in turn reduces overall oxygen consumption rate across the tumor. One cannot rule out that the main effect is simply the result of preferential HT killing of hypoxic tumor cells and that oxygen consumption rate is not important here. However, we argue that oxygen consumption does occur in relatively hypoxic tumor subregions. Hypoxic regions are not totally hypoxic. They are composed of many microscopic foci of hypoxia that also contain well-oxygenated cells near blood vessels [125]. Less hypoxic subregions contain less of these hypoxic foci. Such patterns are readily discernable by looking at the distribution of hypoxia marker drug retention in tumor sections stained immunohistochemically for hypoxia marker drug–protein adducts [126,127]. Killing of aerobic cells lying within relatively hypoxic subregions would contribute to reduced oxygen consumption across a whole tumor. Killing of cells could be by direct coagulative necrosis in regions near the T_10_ values, which are at or above 45 °C. On the other hand, moderate temperature thermal killing (T_50_ values of 42–43 °C) could reduce oxidative phosphorylation [87,109] and/or induce apoptosis in aerobic tumor cells, thereby contributing to reduced oxygen consumption as well as reducing tissue pressure to enhance perfusion [128]. However, we acknowledge that further work would be needed to resolve whether direct hypoxic tumor cell killing alone or in combination with reduced oxygen consumption rate contributes to reoxygenation. One method that could be used to resolve this question is ^15^O PET [129].

Importantly, reoxygenation does not occur in all subjects. In fact, hypoxia is exacerbated 24 h post HT in some subjects [104,107,119]. Mechanisms for this heterogeneous response are not currently delineated. It is possible that the microvasculature in some subjects is less mature and more thermally sensitive. Immature microvasculature is devoid of pericyte coverage and lacks strong endothelial cell junction connections. Such microvessels are sensitive to VEGF withdrawal [130] and are more thermally sensitive [131,132,133]. Selective destruction of such vessels by HT would lead to necrosis and hypoxia. Alternatively, induction of hypoxia could occur as a result of vascular steal. Vascular steal has been described as being responsible for reduced perfusion and increased tumor hypoxia in response to some vasoactive drugs, for example. Upon drug treatment, vasodilation of surrounding normal vasculature occurs [134,135]. Tumor vessels, on the other hand, are often devoid of smooth muscle and cannot vasodilate. Vascular steal occurs because of the shift in flow resistance between normal and tumor tissue, which thereby shunts perfusion to the surrounding normal tissue [135]. Arterioles and venules in normal tissue are more thermally resistant than tumor arterioles [53]. This relative difference in thermal resistance to permanent stasis could increase flow in normal tissue at temperatures that cause vascular stasis in tumors. Further work is needed to more fully explain why reoxygenation occurs in some subjects, while in others, hypoxia is exacerbated. In any case, the heterogeneous response of tumors to HT in different subjects points to the need to measure extent of hypoxia before and during HT treatment regimens in order to differentiate those subjects who benefit from HT-induced reoxygenation vs. for which HT is contraindicated. As described earlier, high rates of heating could also contribute to vascular damage and persistent hypoxia [53,54].

It is likely that the characteristics of the temperature distribution and/or tumor location have an important role in the physiologic response to HT in human subjects. Perfusion was measured prior to and immediately after HT in a subject with cervix and rectal cancer, using H_2_^15^O-PET [136]. Increases in perfusion were not observed. There was an increase in water partition coefficient, which the authors speculated could influence oxygen transport. The temperatures achieved were lower than those seen in sarcomas, averaging 40.7 ± 0.6 °C vs. median temperatures of 41–42 °C in sarcomas [121].

It is also important to consider whether HT induced reoxygenation plays a role in immune surveillance. Both HT and radiotherapy are known to enhance immune surveillance by a range of mechanisms [20,137]. However, both hypoxia and lactic acidosis exert a negative influence on the innate and adaptive immune systems [20,28]. Reoxygenation induced by HT, therefore, could be playing an important role in the enhanced anti-tumor effect of thermoradiotherapy. An increase in perfusion along with killing of hypoxic tumor cells could reduce lactate levels (and increase pHe) as well, thereby contributing to enhanced immunity. We have previously shown a direct positive correlation between HT induced increases in perfusion at 24 h post HT and increases in pHe [120]. We did not find a correlation of these changes with local tumor control after thermoradiotherapy to soft tissue sarcomas in dogs, but increases in pHe 24 h post HT were associated with prolonged metastasis free survival. Low baseline pHe was associated with shorter time to metastasis, as well [109]. Perhaps these differences in tumor acidity at baseline or after HT were associated with tumor immunity. Further work needs to be conducted to define underlying mechanisms.

Although the results shown here support underlying mechanisms for reoxygenation following HT, they are limited by lack of spatially registered data. Functional imaging holds potential to uncover how spatially varying thermal doses affect tumor physiologic response. Using MRI, it is possible to acquire temperature distributions, serial measurements of perfusion distribution and ADC distribution in the same tumor. Oxygen sensitive MR imaging methods and/or ^18^F-misonidazole PET imaging [138] could reveal information about the spatial distribution of hypoxia. Using such data, it would then be possible to estimate the efficiency of cell killing across a tumor.

A preliminary effort was conducted to ascertain the efficiency of cytotoxicity following a thermoradiotherapy treatment in a human soft tissue sarcoma, where non-invasive thermometry was used to ascertain the temperature distribution, and radiation treatment planning revealed the spatial distribution of RT dose within the same tumor. Effects of the varied temperature distribution on cell survival were estimated using extensive cytotoxicity data of CHO cells by Loshek, who measured the time dependence of cell killing for 42 °C HT alone, RT alone and the combination [139]. All of the temperature data within the heated volume of the example case were converted to equivalent minutes at 42 °C, using the Sapareto and Dewey CEM formalism [61]. For further details about the methods for determining cell survival, please see Appendix A for further information. The soft tissue sarcoma in the calf of a human patient is depicted in Figure 4. Figure 4A shows the location of the tumor, as imaged by ADC. Figure 4B shows the temperature distribution, measured by proton resonance frequency shift MRI [81]. Figure 4C depicts the radiation dose distribution from treatment planning. The predicted cell kill within each image pixel from a single dose of radiation is in the range of 50% and is uniform within the irradiated volume because the spatial distribution of radiation was set to be uniform by treatment planning (Figure 4E).

The impact of the varied temperature distribution on cell killing (as depicted by −log10 (survival)) shows highly efficient killing in the hottest tumor regions, along with virtually no killing in the cooler regions of the tumor (Figure 4D). The influence of thermoradiosensitization on cell killing is seen in Figure 4F. Careful examination shows enhanced killing efficiency around areas of cell killing by HT alone (Figure 4D). These data reveal interesting insights into the influence of temperature variation on the distribution of cell killing. First, the extent of cell killing is much greater for HT than for a 2 Gy dose of RT alone within the hotter tumor regions. The greatest cell killing, on the order of 5 logs/pixel, occurs in 10–15% of the tumor region. Killing in these hotter regions would be expected to reduce oxygen consumption rate, thereby contributing to reoxygenation in the rest of the tumor hours to days after HT. Second, although thermoradiosensitization is evident, it is not as extensive as one might project, particularly in the cooler regions of the tumor. Even this one example case suggests that more simulations of this type should be considered, especially if information about hypoxia is added.

We have also conducted a series of simulations of tumor control probability [TCP] based on the Loshek data referred to above [139]. We considered the impact of once weekly HT induced radiosensitization (Appendix A and Appendix A) on cell survival and TCP over a six- or seven-week course of conventionally fractionated radiotherapy. Secondly, we considered the impact of a portion of hypoxic tumor cells moving to the aerobic compartment 24 h post HT (Appendix A and Figure 5). These simulations are based on observations that we made in canine sarcomas [119]. Even a 30% shift after each weekly HT leads to a TCP nearing 100%. On the other hand, TCP drops quite significantly if a tumor becomes more hypoxic after HT, as we have observed in some subjects. Lack of reoxygenation is predicted to render the tumor described as incurable with the radiotherapy doses described.

Despite clear evidence that reoxygenation can occur up to 24–48 h after HT in some canine and human tumors, it is not definitively known whether reoxygenation occurring in an individual’s tumor is associated with long-term treatment outcome. We report on two small subset analyses in canine soft tissue sarcomas suggesting that reoxygenation after the first HT can influence duration of local tumor control after thermoradiotherapy. However, validation is required in larger patient series. Future studies should be directed toward answering whether changes in oxygenation after HT correlate with local tumor control and progression free and overall survival. We also caution that the human sarcoma and locally advanced breast cancer and canine sarcoma data reported in the review are based on several small studies. Further clinical trials, with greater numbers of subjects, would be needed for validation of the observation that reoxygenation after HT results in better anti-tumor effect.

It is also important to note that many other factors, independent of reoxygenation or thermal dose, per se, may influence treatment response to thermoradiotherapy. Examples include: (1) technical variations in application of HT [19,78], (2) variations in sequence and/or time interval between HT and radiotherapy [19], (3) rate of heating [54]; other physiologic factors such as pH, perfusion and/or metabolism and patient specific factors such as age [140,141], smoking history [142,143] and genomic variation [112,144]. Thus, as we have attempted to tease out how hypoxia and reoxygenation influence treatment outcome, it is important to keep in mind that many factors can play into the ultimate outcome for a specific patient. Trials conducted in the future could benefit from data acquisition of as many potential mitigators as possible.

## 7. Returning Back to Differences in Temperature Distributions between Rodent and Human Tumors

Finally, we need to come back to our original premise that differences in temperature distribution between rodent tumors vs. human and canine tumors are physiologically important. We show that thermal doses at the higher end of the distribution in human and canine sarcomas create ADC changes that are consistent with induction of necrosis and, ironically, reoxygenation. In contrast, temperatures in the lower end of the distribution are associated with increased perfusion. These physiologic changes are associated with treatment response. Such heterogeneity in physiologic response within human and canine tumors would not have been seen in rodent tumors, where water bath heating yields a fairly uniform temperature distribution. This raises the question, then, of why reoxygenation has been observed in some rodent tumors and not in others after uniform mild temperature water bath heating? There are two potential explanations for this: (1) It has been shown that mild temperature HT increases HIF-1α expression in some tumors [57]. HIF-1, in turn, upregulates PDK-1, which controls the switch from aerobic to anaerobic metabolism. This switch would reduce oxygen consumption rate, thereby contributing to reoxygenation. (2) Mild temperature heating has been reported to induce apoptosis and/or senescence in some tumor cell types in vitro and in vivo [145,146]. The induction of apoptosis and senescence would reduce oxygen consumption rate. Apoptosis could also contribute to improved perfusion as a result of reduced tissue pressure [128]. The preponderance of apoptosis appears to be temperature dependent, with increases occurring with temperature up to 43 °C for 30–40 min [147]; above this, necrosis becomes the primary cell death mechanism [147]. It is likely that the aforementioned putative mechanisms of reoxygenation occur in some tumor lines, but not all. Uncovering mechanisms for variation creates a clear framework for future pre-clinical research, as mechanisms may very well be associated with treatment responses in human tumors as well.

It is also important to consider potential reasons for variation in treatment response, within specific tumor lines. Examination of individual variability in tumor response has rarely been examined in pre-clinical models. One example is provided that involved HT. Palmer et al. examined individual responses of the ovarian tumor model, SKOV-3, to a thermosensitive liposome containing doxorubicin [148]. The tumors were heated to 42 °C for 60 min by water bath. Using optical spectroscopy, they measured hemoglobin saturation [Hb_sat_], total hemoglobin and drug concentration in heated tumors. The primary outcome variable was growth time [time to reach 3 times treatment volume]. Hb_sat_ and drug concentration were significantly related to growth time. Further, cluster analysis revealed that tumors with both low Hb_sat_ and low total Hb had relatively short growth times. Total Hb is related to blood volume and perfusion rate. Although optical spectroscopy is not widely available, there are many other ways to non-invasively measure parameters related to tumor hypoxia, perfusion and ADC in mice, using MRI or PET [24,149]. It is recommended that pre-clinical study designs involving monitoring of individual treatment responses be considered for future research. Additionally, it is advised to use heating methods that yield peaked temperature distributions that mirror what is seen clinically. For example, Kelleher used a near infrared method that achieved a peaked temperature distribution in a rat tumor line [84]. Such studies could prove invaluable in setting the stage for future human clinical trial designs.

## 8. Conclusions

In this review, we provide convincing evidence that HT causes prolonged reoxygenation lasting at least 24–48 h in both human and canine cancers. Further, we show that reoxygenation is likely caused by increased perfusion as well as a putative reduction in oxygen consumption rate. Importantly, these effects are linked to characteristics of the peaked temperature distribution that usually accompany HT treatment of solid cancers in the clinic. The higher end of the temperature distribution is associated with evidence of cell killing and/or reduced oxygen consumption rate, whereas temperatures at the lower end of the distribution are associated with increases in perfusion. These effects appear to be occurring simultaneously in tumors after HT.

We hypothesize that the relative lack of validation of such results in pre-clinical models is due to the fact that rodent tumor heating is usually performed in water baths that do not yield peaked temperature distributions seen in the clinic.

Finally, we end with a suggestion for future clinical studies that carefully examine the impact of HT on cell killing and physiology by combining functional imaging with estimates of cell survival based on in vitro cell survival curve data. Such studies are likely to provide important insights into which features of HT+RT (direct cell killing by HT, direct cell killing by RT, reoxygenation influence on RT cell killing and heat radiosensitization) will have the greatest influence on local tumor control.

## Figures and Tables

**Figure 1 cancers-14-01701-f001:**
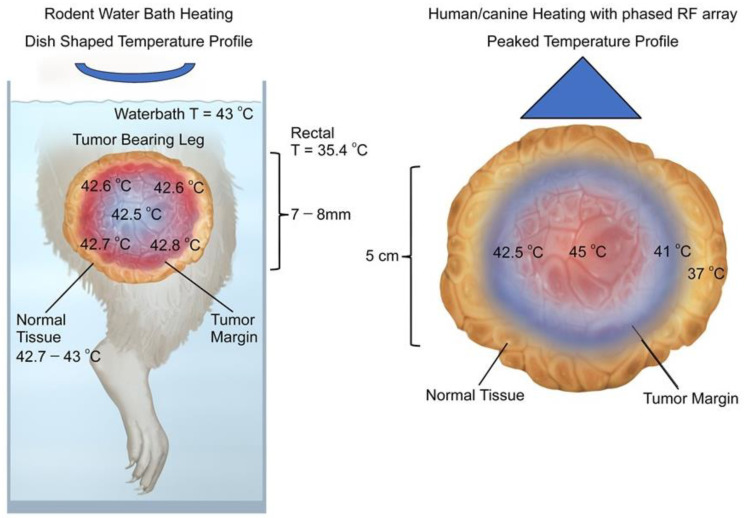
Schematic comparison of temperature distributions for rodent water bath heating vs. temperatures seen in canine and human tumors heated with radiofrequency or microwave devices. (**Left Panel**) Temperature distributions in rodent tumors heated with water baths tend to be relatively uniform [profiles are dish-shaped], with highest temperatures at the margin of the tumor, while intra-tumoral temperatures are slightly cooler and relatively uniform. Depicted data are taken from a paper by O’Hara et al., where detailed intra-tumoral temperatures were documented using micro-thermocouples [72]. Although not shown in color for clarity, the whole leg is at elevated temperature. This is described numerically at the left side of the figure. (**Right Panel**) Temperature distributions in human and canine sarcomas heated with phased radiofrequency devices have a peaked temperature distribution in which the temperatures closer to the center are higher than those at the tumor edge. Typically, some surrounding normal tissue is heated to mild temperatures, as depicted. Note also that maximum intra-tumoral temperatures are higher than what is seen in rodent tumors. This is a schematic representation of non-invasive thermometry obtained in human sarcomas [79,80,81].

**Figure 2 cancers-14-01701-f002:**
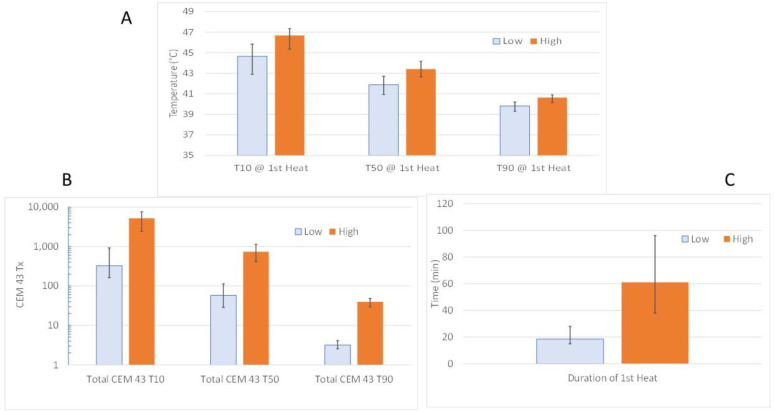
Thermal dose characteristics of thermal dose escalation trial of Thrall et al. [108] (**A**) Comparison of CEM43T_10_, T_50_ and T_90_ values. The trial was designed to deliver approximately a 20-fold difference in CEM43T_90_ between the low and high dose groups. (**B**) To achieve this difference in CEM43T_90_, and CEM43T_10_, T_50_ throughout the tumors were higher. (**C**) Heating times were also longer. *N* = 21 Low dose group; *N* = 18 High dose group. Data from the subjects for which physiologic data were reported (Lora-Michaels et al. [109]).

**Figure 4 cancers-14-01701-f004:**
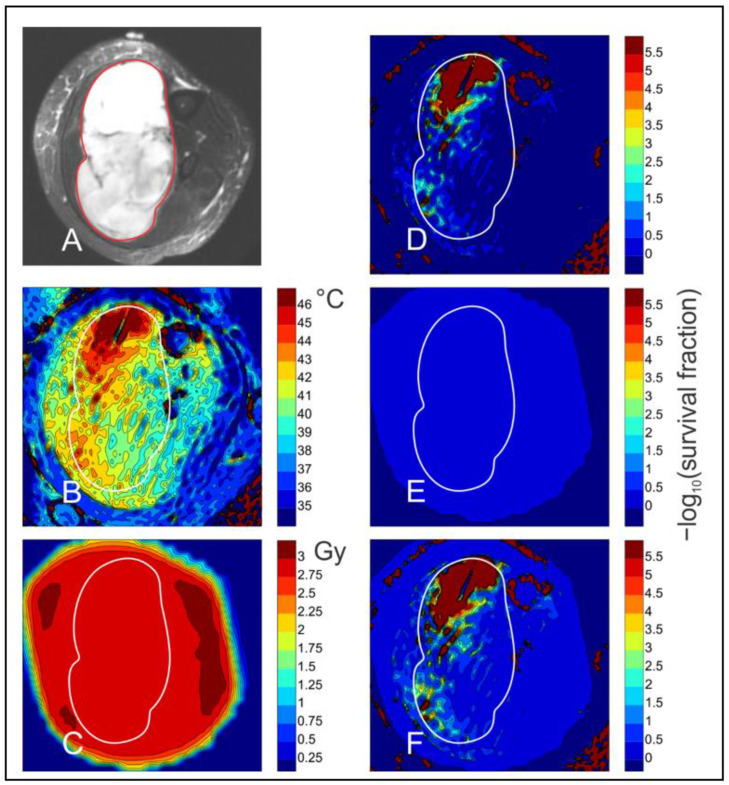
Imaging and simulation of combined HT and radiation treatment of a sarcoma. Images show a cross-section through a human patient’s calf. Simulations are based on results of Loshek et al. [139] for dependence of survival fraction of Chinese hamster ovary cells on doses of combined radiation and heating, together with results of Sapareto and Dewey [61] for the dependence of thermal dose on temperature. The period of heating was 54 min. Details of the simulation are provided in Appendix A. (**A**) Diffusion weighted MRI image of thigh cross-section. Tumor region is outlined in red and transferred to other images. (**B**) Temperature distribution in tissue during hyperthermia, obtained by non-invasive MRI thermometry. (**C**)Radiation dose derived from treatment plan. (**D**) Predicted cell kill from HT alone. All cell kill values are expressed in terms of −log10 (survival fraction). (**E**) Predicted cell kill from radiation alone. (**F**) Predicted cell kill from combined HT and radiation.

**Figure 5 cancers-14-01701-f005:**
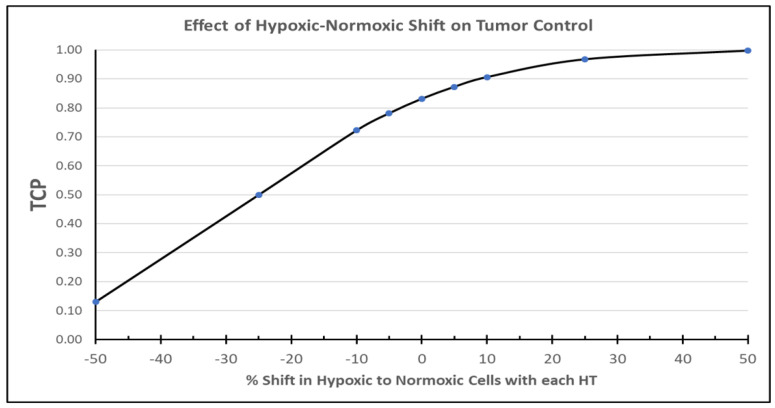
Predicted tumor control probability [TCP] for conventionally fractionated radiotherapy + HT, where HT is administered once weekly. The impact of HT induced reoxygenation at 24 h post HT is depicted, as the proportion of hypoxic cells that reoxygenate vs. proportion of aerobic cells that become more hypoxic after each HT. TCP reaches nearly 100% if even 30% of hypoxic tumor cells reoxygenate 24 h after each HT, thereby affecting cytotoxicity of the radiotherapy fraction given the day after HT. On the other hand, TCP drops quickly in a condition where aerobic tumor cells become more hypoxic 24 h after HT. Although reoxygenation occurs more frequently with HT, one must remain aware of the smaller population of tumors that become more hypoxic after HT, as such effects are predicted to substantially reduce TCP. Details of these simulations are shown in Appendix A.

**Table 1 cancers-14-01701-t001:** Physiological Predictors of Time to Local Failure: Thermal Dose Escalation Trial.

Variable	Parameter Estimate	Hazard Ratio	Score *p*-Value	Wald *p*-Value
HF Post-Pre	−0.0643	0.94	0.0070	0.0340
Median pO2 Post-Pre	0.0896	1.09	0.0230	0.0710
Pimo % area	−6.549	0.00	0.038	0.0510
PDE/ATP	0.2246	1.25	0.0490	0.0640

HF = Hypoxic Fraction—fraction of measurements <10 mmHg; *N* = 11 for HF and Median pO2; *N* = 16 Pimo area; *N* = 13 PDE/ATP; Pimo = pimonidazole; (PDE/ATP data previously published, Lora-Michaels et al. [109]).

**Table 2 cancers-14-01701-t002:** CEM43Tx vs. change in volume, ADC, pO2, HF, iAUC: Thermal Dose Fractionation Trial.

Variable	*N*	CEM43T_10_		CEM43T_50_		CEM43T_90_	
		Coefficient	*p*-Value	Coefficient	*p*-Value	Coefficient	*p*-Value
Change ADC Pre-post *	29	−0.53	0.0030	−0.56	0.0015	0.11	0.5665
iAUC change 24 h ^	17	0.07	0.7798	0.23	0.3599	0.5109	0.0311
Median pO2 change at 24 h ^	38	0.38	0.0214	0.27	0.1087	−0.07	0.9829
Change HF 24 h ^	38	−0.34	0.0451	−0.27	0.1074	−0.07	0.674
Volume change Pre-Post *	38	−0.42	0.0084	−0.36	0.0258	0.2983	0.17

* Total CEM43Tx; ^ CEM43Tx first HT; HF = % measurements < 10mmHg; iAUC = DCE-MRI perfusion parameter; HF = hypoxic fraction.

**Table 3 cancers-14-01701-t003:** Physiological Predictors of Tumor Volume Change: Thermal Dose Fractionation Trial.

Variable	*N*	Coefficient	*p*-Value
iAUC median change 24 h	17	−0.47	0.0472
Median pO2 change 24 h	38	−0.040	0.0146

iAUC = DCE-MRI perfusion parameter.

## Data Availability

Previously unpublished data presented in this paper can be provided upon request.

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
