# Peer review of "Accurate Three-Dimensional Thermal Dosimetry and Assessment of Physiologic Response Are Essential for Optimizing Thermoradiotherapy"

_cancers, 2022, doi:10.3390/cancers14071701_

Round 1

Reviewer 1 Report

This manuscript by Dewhirst et al addresses an important topic and gives an excellent overview of the topic, including not just the evidence but also the gaps which need to be filled. I recommend publication, subject to addressing some minor issues.

The abstract gives a good summary, but at the same time I feel the content is perhaps too qualitative, the abstract lacks almost all quantitative detail. I urge the authors to revisit the text of the abstract and check whether the present abstract text really reflects the content of the paper.

the introduction is compact and a good prelude to the main theme of the manuscript. References may not always be the top reference, I would for instance add in line 61 this reference:

Overgaard J. Hypoxic modification of radiotherapy in squamous cell carcinoma of the head and neck--a systematic review and meta-analysis. Radiother Oncol. 2011 Jul;100(1):22-3

the introduction implicitly suggests that effects of hyperthermia are limited to effects on the hypoxic fraction. I understand that most space is dedicated to effects on hypoxia as this is the topic of this paper, but I would suggest to also (briefly) mention the wider cytotoxic effects of hyperthermia in the introduction and/or the discussion, with references.

The authors could also discuss the fact that radiosensitization appears to increase significantly with shorter time between radiotherapy and hyperthermia, and perhaps link that to the different hyperthermia effects they discuss.

One caveat to add stronger is that the evidence rests on a fairly small number of trials, and that many events can be explained in different ways, resulting in a chicken or egg problem. , e.g. no clear conclusions can be drawn when a high CEM43T10 is associated with the largest change in pO2. Perhaps CEM43T10 is high when there is an ill-perfused hypoxic subsection with little heat removal, meaning that that subvolume heats easily and thus responds better, resulting in a larger reduction of hypoxia.

revisiting the earlier trials with new information from those trials is very interesting with intriguing data. I would also suggest converting (part of) the tables to bar charts which may more readily summarize the findings for the reader in one glance

Author Response

Reviewer #1.  We thank the reviewer for the excellent suggestions and complementary comments about this review.  Our specific responses are listed below.

  1. Add Overgaard’s review on hypoxic modifications in head and neck cancer. This paper was added.
  2. The reviewer asked that we mention some of the other effects of HT that are relevant to its use with radiotherapy. Material summarizing some of these key features was added to the Introduction, See lines 67-87.

Here is the specific text:

HT induces a number of biologic and physiologic effects on tumors. HT inhibits multiple DNA damage repair mechanisms, which play a major role in heat radiosensitization. The inhibition of DNA repair provides rationale for combining HT with HSP90 (heat shock protein-90) and/or PARP (poly (ADP-ribose) polymerase) inhibitors[6].  Heat shock proteins, HSP70 and HSP27 bind to enzymes to facilitate base excision repair[7].  This heat shock protein association may enhance DNA damage repair after HT. Substantiating this hypothesis is the observation that enhancement of repair of heat induced double strand breaks is linked to HSP70 and HSP27 association with heat labile DNA polymerase beta in thermotolerant cells[8].  The thermotolerance-induced enhancement of DNA damage repair could reduce effectiveness of radiotherapy treatments given when cells are thermotolerant [9, 10].  If so, such an effect could reduce the impact of reoxygenation observed 24-48h post HT, which is the main subject of this review. It is unknown whether this mechanism of thermotolerance induced radioresistance is clinically relevant. Further research would be needed to answer this question.

Hyperthermia also induces a number of immunostimulatory effects in both the innate and adaptive immune systems [11] that are likely important for its biological effectiveness when combined with radiotherapy. HT is cytotoxic itself, with the extent of cytotoxicity being dependent upon the time and temperature of heating[12]. Further, the cytotoxicity of HT is not dependent upon oxygen availability, so it is complementary to radiation in this respect, since hypoxia causes significant reduction in cytotoxicity of radiotherapy [13].

3. In the next paragraph we also briefly discuss the fact that maximal radiosensitization occurs when HT and RT are given in close proximity to each other. Lines 88-90

4. The reviewer also asked us to discuss optional explanations for the observation that T10 is associated with reoxygenation. The suggestion is that high T10 values occur because of poor perfusion, thereby selectively killing hypoxic tumor cells.  We agree with this explanation to some extent.  But we take this explanation further.  We are arguing that the cytotoxicity that occurs at higher temperatures reduces the oxygen consumption rate across the tumor and that this reduction in oxygen consumption rate is at least partly responsible for reoxygenation. We have attempted to make this explanation clearer in the discussion – see 2nd paragraph under Section 6. This idea must be predicated on the fact that oxygen consumption occurs even in regions that have more predominant hypoxia.  Examination of many papers using hypoxia marker drugs reveal that even very hypoxic regions contain well-oxygenated cells located near blood vessels. See lines 547-565

Here is our specific response to this question:

We have speculated that reoxygenation occurs as a result of direct HT cytotoxicity of aerobic cells, which in turn reduces overall oxygen consumption rate across the tumor.  One cannot rule out that the main effect is simply the result of preferential HT killing of hypoxic tumor cells and that oxygen consumption rate is not important here. However, we argue that oxygen consumption does occur in relatively hypoxic tumor subregions.  Hypoxic regions are not totally hypoxic. They are composed of many microscopic foci of hypoxia that also contain well oxygenated cells near blood vessels [115]. Less hypoxic subregions contain less of these hypoxic foci.  Such patterns are readily discernable by looking at the distribution of hypoxia marker drug retention in tumor sections stained immunohistochemically for hypoxia marker drug-protein adducts[116, 117].  Killing of aerobic cells lying within relatively hypoxic subregions would contribute to reduced oxygen consumption across a whole tumor.  Killing of cells could be by direct coagulative necrosis in regions near the T10 values, which are at or above 45°C. On the other hand, moderate temperature thermal killing (T50 values of 42-43°C) could induce apoptosis in aerobic tumor cells, thereby contributing to reduced oxygen consumption as well as reducing tissue pressure to enhance perfusion[118].  However, we acknowledge that further work would be needed to resolve whether direct hypoxic tumor cell killing alone or in combination with reduced oxygen consumption rate contributes to reoxygenation. One method that could be used to resolve this question is 15O PET[119].

5. The reviewer also challenged us to be cautious about over-interpreting our results, given that they are based on a small number of clinical trials. We added the following text to Section 6, See lines 690-693

“We also caution that the human sarcoma and locally advanced breast cancer and canine sarcoma data reported in the review are based on seven small studies. Further clinical trials, with greater numbers of subjects would be needed for validation of the observation that reoxygenation after HT results in better anti-tumor effect.”

6. The reviewer asked that we plot some of the tabular data. We have plotted up the thermal data from the canine thermal dose escalation trial. The T10, T50 and T90 data for the Thermal dose equivalence trial were very similar – we moved the tabular data from that trial to supplemental data.

Reviewer 2 Report

The manuscript focusses on the influence of re-oxygenation onto HT-RT treatment outcome as a process leading to an improved tumour response. Regarding physiological short-term up-regulation of perfusion and the combination of a few HT sessions with RT fractions (fx), one would not expect a large increase of radio-sensitization due to re-oxygenation. Nevertheless, the manuscript provides very interesting information and explanations why perfusion-related effects may be important. Especially the provided evidence for a long-term effect (increased oxygenation after 24-48 h) may be an important explanation for the impact of HT-induced perfusion changes onto tumour response. It would be interesting to have some more quantitative data regarding for example TCP’s for a whole treatment course consisting of e.g. 32 fx RT combined with a few (let’s say 3-6) HT sessions leading to a perfusion-mediated increase of radio-sensitivity (with an OER of typically 2-3).

In general, the paper provides a very interesting collection of important points and can be seen as a valuable contribution to the actual discussion regarding the different, potential key processes responsible for therapy outcome of HT-RT treatments. I fully agree with the conclusion that functional imaging should be applied more systematically to elucidate the influence and importance of perfusion (and perfusion related processes which should not only focus on re-oxygenation). In this regard, the title of the manuscript may be modified, since not only thermal dosimetry by measuring temperatures and calculating CEM-values is recommended. “Thermal doses” should be linked to monitored, spatio-temporal effects in the tumour volume.

Specific points:

(1) p5, paragraph 4.2; line 212: Direct cell killing of 30-50% @ T90 = 41°C - what are the reported max Temperature?

(2) Sect. 5.2 p6., 1st paragraph, line 239: “several papers were published, verifying his hypothesis” Regarding the fact that apart from direct cell killing and perfusion-mediated re-oxygenation, several other mechanisms may be involved, Oleson’s hypothesis was not really verified by these papers (this would be very difficult in a strictly scientific manner). The mentioned papers rather support some evidence for a long term effect of perfusion.

(3) Chapt. 6, p12., 3rd paragraph, line 473: Role of re-oxygenation in immune surveillance – assuming that oxygenation is associated with enhanced perfusion, why is oxygenation the key factor and for example not the wash-out of acidic metabolites or immune cell accessibility?

Missing points in the Discussion:

(4) The conclusions are – corresponding to the focus of the paper – based on the discussion of the role of oxygenation vs. direct cell killing. In this somehow limited view, the statement that lower temperatures are associated with perfusion-related effects – which is mainly considered to be effective via re-oxygenation - and higher temperature with direct cell killing is obvious. Nevertheless, beside direct cell killing at higher temperatures, synergistic effects between RT and HT may contribute significantly to the response – it would be interesting to have at least some explanation regarding the relationship between the different processes (including the point that perfusion does not only increase re-oxygenation and therefore, other perfusion-related effects may correlate indirectly with pO2 levels?). In this light, the concluding sentence about the importance of combining functional imaging with estimates for cell survival in vitro (HT-induced cell death processes on cellular level) is certainly a very important point.

(5) Why has RT prior to HT at mild temperatures an effect – regarding oxygen-mediated radio-sensitization, what is the expected impact of re-oxygenation for this sequence?

Author Response

Reviewer #2 We thank Reviewer #2 for the insightful and helpful suggestions.

  1. The reviewer asked for clarification of what the distribution of higher temperatures were in the theoretical paper by Rosner et al, where a T90 was associated with 30-50% cell killing. Since the results were based on multiple simulations (10 different randomly generated tumor geometries, 3 different T90 values, 2 different perfusion values and 3 different applied powers), there is not one value to assign to descriptors of the higher end of the distribution.  However, the data used to create the simulations included cell survival data for temperatures ranging from 41.5-45.5°C, so the simulations were constrained to not exceed temperatures of 45.5°  We added a brief explanation for this. See Lines 278-280

Here is the added text:

“This occurs because of cell killing temperatures higher than the T90.  Simulated temperatures above the T90 ranged up to 45.5°C. Thermal killing of 30-50% of tumor cells would be sufficient to have an important impact on oxygen consumption rate and tumor hypoxia[41]”

2. The reviewer takes issue with our statement that papers were published verifying Oleson’s hypothesis that reoxygenation after HT must play a role in HT radiosensitization. We softened the language to read “Subsequent to Oleson’s paper, several papers were published, showing results that are consistent with his hypothesis.” See line 304

3. The reviewer asked about the role of pH in regulating immune surveillance. We added a short discussion about this point.  First, we have written some reviews on this subject, which we refer to in the discussion. Basically, low pH, high lactate and hypoxia contribute to suppression of immune surveillance by different mechanisms.  Fortuitously, we actually looked at whether extracellular pH (pHe) at baseline and changes after the 1st HT were related to treatment outcome in dogs with soft tissue sarcomas that underwent thermoradiotherapy.  Here is the text that we added on this point. See lines 595-609

“It is also important to consider whether HT induced reoxygenation plays a role in immune surveillance. Both HT and radiotherapy are known to enhance immune surveillance by a range of mechanisms[11, 22, 127]. However, both hypoxia and lactic acidosis exert a negative influence on the innate and adaptive immune systems[11, 22]. Reoxygenation induced by HT, therefore, could be playing an important role in the enhanced anti-tumor effect of thermoradiotherapy. An increase in perfusion along with killing of hypoxic tumor cells could reduce lactate levels (and increase pHe) as well, thereby contributing to enhanced immunity.  We have previously shown a direct positive correlation between HT induced changes in perfusion at 24h post HT and increases in pH[109].  We did not find a correlation of these changes with local tumor control after thermoradiotherapy to soft tissue sarcomas in dogs, but increases in pHe 24h post HT were associated with prolonged metastasis free survival. Low baseline pH was associated with shorter time to metastasis as well [100]. Perhaps these differences in tumor acidity at baseline or after HT were associated with tumor immunity. Further work needs to be done to define underlying mechanisms.”

4. The next comment is not very clear to us, but we think that the reviewer is asking whether other mechanisms of HT radiosensitization might be playing a role in the enhanced anti-tumor effects associated with reoxygenation. We already brought up the idea that enhanced anti-tumor immunity might be important and that this might be related to both reoxygenation and alkalinization in the discussion.  In the introduction, we also brought up the importance of inhibition of DNA damage repair by HT.  We added a paragraph about this, at the request of reviewer #1.  We do not believe that we need to provide additional explanation, but the complexity of events that occur, clearly points to the need for multiparametric imaging, as the reviewer astutely points out. See lines 67-86

5. The reviewer asked about how sequencing between HT and RT might play a role in the reoxygenation effect. We have not examined this question, nor has it been examined by any other investigators, to our knowledge.  But it is important to remember that the reoxygenation that we are describing is occurring 24-48h after HT, so any reoxygenation that occurs during or shortly after HT is really irrelevant to our results. We did not comment about sequencing in the paper.

6. We changed the title of the paper to emphasize the idea of obtaining spatially related dosimetry and physiologic data

7. In the general comments section, the reviewer suggested that we might wish to do some TCP calculations.  The underlying question would be whether our observation of reoxygenation 24h after HT would be sufficient to impact TCP. We have added some calculations showing that this reoxygenation would be sufficient to impact TCP. See lines 660-669 and Supplemental data

Reviewer 3 Report

The article deals with an actual topic of hyperthermia combination with radiotherapy. Such a comprehensive collection of our current knowledge is a gap-filling. The review is well organized, clear, and well readable.

I have some recommendations to complete the overview:

The paper deals with hyperthermia as a unified treatment with all heating devices, characterized by only the  dose. This interpretation is incorrect. The whole-body hyperthermia (WBH) has perfect homogenous heating and high thermal  dose, but its response rate is not as high as many local treatments which lower doses. The device differences appear in local heating as well. For example, the controversial clinical observation of the thermoradiotherapy protocols of [[1]] and [[2]], or the apparent controversy of human cervix results between [[3]] and [[4]] show the challenge. The device's ability to heat quickly or slowly also affects the blood perfusion rate [[5]]. The review has to remark the differences, which need further research.

Together with oxygenation, the DNA repairing enzymes also have a crucial role in the success of radiation therapy. The growing temperature increases the enzyme processes, which opposites the grooving oxygen content to keep the damages of DNA strands. So these effects are balanced. The hyperthermia developed HSP70 and HSP27 regulate the base excision repair enzymes in response to stress [[6]]. The "warming up" heating period appears as a preheating, which could increase the activity of reparation enzymes [[7]]. The review has to mention the enzymatic effects as counterbalancing of the oxygenation processes.

Increasing the oxygen content by switching the anaerobic effects is a double-edged sword. The intensive production of pyruvate decreases the pH of the tumor, which increases the formation of metastases [[8]]. The low pH independently from hypoxia induces the spread of tumor cells, worsening tumor patients' long-term prognosis. This factor also has a place in the review.

Additionally, the temperature increase can produce vasoconstriction in certain tumors, which decreases the BF and the decrease in heat exchange offers a relatively higher temperature in these regions [[9]]. This effective heat trap [[10]] lowers the available oxygen, affecting the efficacy of RT. The  growths by the increasing temperature, but in temperature 42.5 it could turn back [[11]]. The posttreatment value of  also changes, and its maximum has an optimal temperature [[12]].

I have some suggestions to complete the review:

  1. 2/76-77. Definition of the hyperthermic temperature intervals varies in the literature. Hannon describes the mild local hyperthermia as between 40-42, while the moderate interval 42-45[[13]]. Obviously, the whole-body hyperthermia (WBH) does not use the 43upper limit of the mild category. WBH defines the temperature 38<T<41, and 41<T<42 as "extreme" [[14]], [[15]]. I propose that the authors have to mention the discrepancies in their definition. 

Fig.1. p. 3-4/128-130. The figure misleads the reader. It shows a definite and rigid temperature barrier at the tumor margin. The temperature does not stop at the tumor border, spreading over time.

  1. 5/185-187. The reference [64] shows the evident upregulation of the HIF-1with mild hyperthermia 4130 min, in vivo. The correct results are: "Irradiation of FSaII tumours with a single dose of 15Gy led to significantly decreased bloodperfusion, increased hypoxia and upregulation of HIF-1a and VEGF. On the other hand, MTH at 41 _C for 30 min increased blood perfusion and tumour oxygenation, thereby suppressing radiation-induced HIF-1a and VEGF in tumours, leading to enhanced apoptosis of tumour cells and tumour growth delay." Your reference is incorrect, but the final evaluation (p.5/187) is appropriate.

p.5/201. It is a mistake to use "Arrhenius theory". Arrhenius theory refers to the acid-base composition. The Arrhenius law (or Arrhenius plot) is the rate of chemical reaction depending on temperature. It is an empirical law. Eyring had a correct quantum-mechanical theory, unlike Arrhenius (Eyring theory).

References

[1]          Kroesen M, Mulder HT, van Holthe JML, et.al. (2019) The effect of the time interval between radiation and hyperthermia on clinical outcome in 400 locally advances cervical carcinoma, Frontiers in Oncology, 9:134, doi: 10.3389/fonc.2019.00134

[2]          Crezee H, Kok HP, Oel AL, et.al. (2019) The impact of the thime invterval between radiation and hyperthermia on clinical outcome in patients with locally advanced cervical cancer, Frontiers in Oncology, 9:412, doi: 10.3389/fonc.2019.00412

[3]          Vasanthan, A., Mitsumori, M., Part, J.H. et. al.: Regional hyperthermia combined with radiotherapy for uterine cervical cancers: a multiinstitutional prospective randomized trial of the international atomic energy agency. Int. J. Rad. Oncol. Biol. Phys. 61, 145-153 (2005)

[4]          van der Zee J, Gonzalez Gonzalez D, van Rhoon GC, van Dijk JD, van Putten WL, Hart AA. 2000. Comparison of radiotherapy alone with radiotherapy plus hyperthermia in locally advanced pelvic tumors: a prospective, randomised, multicentre trial. Dutch Deep Hyperthermia Group., The Lancet; 355(9210):1119–1125.

[5].         Hasegawa T, Gu Y-H, Takahashi T, Hasegawa T, Yamamoto IY. (2003) Enhancement of Hyperthermic effects using rapid hyperthermia, In: Kosaka M, Sugahara T, Schmidt KL, Simon E. (eds.) Theoretical and experimental basis of Hyperthermia. Thermotherapy for Neoplasia, Inflammation, and Pain, Springer Verlag Tokyo, pp. 439–444

[6].         Mendez, F.; Sandigursky, M.; Franklin, W.A.; et.al. Heat-shock proteins associated with base excision repair enzymes in HeLa cells, Radiation Research, 153, 186-195.

[7].         Daniel, R.M.; Danson, M.J. Temperature and the catalytic activity of enzymes: A fresh understanding, FEBS Letters, 2013, 587(17): 2738-2743.

[8].          Donato C, Kunz L, Castro-Giner F, et.al. (2020) Hypoxia triggers the intravasation of clustered circulation tumor-cells, Cell Rep., 32(10):108105, doi: 10.1016/j.celrep.2020.108105

[9]          Vaupel, P.; Kallinowski, F.; Okunieff, P. Blood flow, oxygen and nutrient supply, and microenvironment of human tumours: a review. Cancer Res, 1989, 49(23):6449–6465.

[10].       Takana, Y. Thermal responses of microcirculation and modification of tumour blood flow in treating the tumours. In: Kosaka, M.; Sugahara, T.; Schmidt, K.L.; Simon, E.; (eds.) Theoretical and experimental basis of Hyperthermia. Thermotherapy for Neoplasia, Inflammation, and Pain, Springer Verlag, Tokyo, 2001, pp. 408–419.

[11].       Vujaskovic, Z.; Song, C.W. Physiological mechanisms underlying heat-induced radiosensitization. International Journal of Hyperthermia, 2004, 20:163–174.

[12].       Song, C.W.; Shakil, A.; Osborn, J.L.; Iwata, K. Tumour oxygenation is increased by hyperthermia at mild temperatures. Int. J. Hyperthermia, 2009, 25:91–95.

[13].        Hannon G, Tansi FL, Hilger I. (2021) The effects of localized heat on the hallmarks of cancer, Advanced Therapeutics, 4, 2000267, 1-23

[14].        Von Ardenne A, Wehner H. (2013) Extreme whole-body hyperthermia with water-filtered infrared-A radiation, Madame Curie Bioscience Database, https://www.ncbi.nlm.nih.gov/books/NBK6551/

[15].        Hildebrandt B, Hegewisch-Becker S, Kerner T, et.al. (2005) Current status of radiant whole body hyperthermia at temperatures >41.5°C and practical guidelines for the treatment of adults. The German ‘Interdisciplinary Working Group on Hyperthermia’, International Journal of Hyperthermia, 21:2, 169-183

Author Response

Reviewer #3.

  1. The reviewer points out that use of whole-body hyperthermia induces different physiologic effects, compared with methods for inducing local-regional heating. We cannot get into a discussion of whole-body heating effects in this review.  The subject of total body hyperthermia would require a completely different review, which would be timely, but far too extensive and off the subject of this paper.  To make this point clear, we have amended the first sentence of the Introduction to read “Dozens of randomized trials conducted over the past 35 years have shown therapeutic benefit of local and regional HT, when combined with radiotherapy and/or chemotherapy.” See line 58
  2. In the same paragraph, the reviewer attracts our attention to a paper by Kroesen and a commentary by Crezee, dealing with the subject of whether time interval between radiotherapy and HT, when given on the SAME day (within 4h of each other) is important in treatment outcome. We appreciate the reviewer for letting us know about this controversy, but time interval between treatments on the SAME day is not relevant to our paper, where we are focusing on how reoxygenation 24-48h after HT affects treatment response.  We decline to discuss this point in this paper, as it is off subject. Similarly, discussion of clinical trial results using thermoradiotherapy for locally advanced cervix cancer is not relevant to this paper.
  3. The reviewer also points out that rate of heating may influence treatment outcome. We were unable to obtain the paper that the reviewer is referring to, but we have added text discussing the influence of rate of heating on physiologic response – ironically, work published by our group in 1984.  In the work described in this paper, it is not possible to obtain rate of heating data, so it is not known whether the physiologic responses observed were related to that. We point this out as a limiting factor. See lines 108 and 116
  4. The reviewer has asked that we add some information about how HT affects DNA damage repair. Reviewer #1 also asked for this. We added a brief discussion in the Introduction.  The effect of HT on enzyme kinetics is not relevant to this review in our opinion, since we are focusing on how HT reoxygenates tumors 24-48h after  HT. We did add some text about the association of HSP’s with enzymes involved in base excision repair and how that might reduce the impact of reoxygenation 24-48h post HT. See lines 67-79
  5. The reviewer also asked that we include discussion on how HT might affect pH and whether this has any prognostic effects. Reviewer #2 also asked about this. We added a brief discussion on this point and refer to our prior paper by Lora Michaels et al, where we demonstrated that metastasis free survival of dogs with acidic soft tissue sarcomas had significantly shorter metastasis free survival. Additionally, we showed benefit of perfusion mediated alkalinization 24h after the 1st HT on these same subjects.  Alkalinization was significantly associated with increased perfusion after HT.  In keeping with the reviewer’s concerns, metastasis free survival was shorter for those tumors that acidified – this was linked to lack of an increase in perfusion. See lines 595-608
  6. The reviewer suggests that HT can produce vasoconstriction in some tumors. We respectfully disagree with this assertion. We studied arteriolar responses to HT using window chamber tumors, where the tumor feeding arterioles were identified.  We saw no evidence for vasoconstriction. In most cases, these vessels either vasodilated or were unchanged by temperature.  There are two other potential mechanisms that could explain long term changes in perfusion 24-48h post HT; namely direct microvascular damage leading to stasis and/or vascular steal.  If vascular steal occurs, this would reduce tumor blood flow and could be interpreted as vasoconstriction.  We added a discussion on this point. See lines 566-586
  7. The reviewer has asked that we reference the work by Hannon et al, who also defined temperature ranges for “mild”, “moderate” and “irreversible” thermal damage. We have added that to the references, but also point out to the reviewer that the rationale for choosing these ranges was not explicitly described by these authors.  We adjusted our ranges to fit more closely to Hannon et al, but have a short explanation for why we have chosen these descriptors. See lines 118-132
  8. The reviewer has asked that Figure 1 be redrawn. We have worked with a commercial artist to improve the appearance as suggested.
  9. We corrected the description of results from reference 64, as requested. See lines 240-250
  10. The reviewer takes issue with referring to Arrhenius theory, which was the basis for the Sapareto and Dewey CEM43°C formulation. We agree with this claim and have taken out reference to Arrhenius.  See lines 263-269

Round 2

Reviewer 3 Report

I accept most of the answers, corrections, and extensions of the text. However, the authors' response misses my points of the differences in hyperthermia methods.

I argue that the submitted paper mistakenly generalizes oncologic hyperthermia. The paper deals with hyperthermia as a unified treatment with all heating techniques, characterized by only the  dose. My review focused on the differences between the applied heating methods, which contradicts the published facts. The technical variation sometimes significantly deviates the results, as I mentioned between the WBH and local treatments characterized by the same  dose. The dose determining Arrhenius parameters depends on the heating method, too [[i]]. Some results significantly differ between the variants of locoregional techniques. The difference between the AMC4 [[ii]] and BSD2000 [[iii]] ([[iv]]), as well as the contradictory results of RF8 [[v]] and BSD2000 [[vi]], demonstrate my argument.

The speedy and slow heating influences the blood perfusion and the antitumor effect [[vii]]. (I send you attached the unidentified reference from the book "Thermotherapy for neoplasia, inflammation, and pain. Eds. Kosaka et al. [Springer]"for your files.) The DNA reparative processes vary by the temperature rise, considered a preheating [[viii]], intensifying the reaction rate of the DNA repair enzymes. The heating rate varies by applied techniques, developing an equilibrium between the DNA strand breaks and the repairing activity. Consequently, the technical differences appear again as modification factors.

I respectfully ask the authors to correct the points above in their manuscript before resubmission.

[i]           Whitney J, Carswell W, Rylander N. (2013) Arrhenius parameter determination as a function of heating method and cellular microenvironment based on spatial cell viability analysis, Int J Hyperthermia, 29:4, 281-295, DOI: 10.3109/02656736.2013.802375

[ii]          Kroesen M, Mulder HT, van Holthe JML, et.al. (2019) The effect of the time interval between radiation and hyperthermia on clinical outcome in 400 locally advances cervical carcinoma, Frontiers in Oncology, 9:134, doi: 10.3389/fonc.2019.00134

[iii]         Van Leuwen CM, Oeii AL, Chin KWTK, et.al. (2017) A short time interval between radiotherapy and hyperthermia reduces in-field recurrence and mortality in women with advanced cervical cancer, Radiation Oncology, 12:75, DOI 10.1186/s13014-017-0813-0

[iv]         Crezee H, Kok HP, Oel AL, et.al. (2019) The impact of the thime invterval between radiation and hyperthermia on clinical outcome in patients with locally advanced cervical cancer, Frontiers in Oncology, 9:412, doi: 10.3389/fonc.2019.00412

[v]          Vasanthan, A., Mitsumori, M., Part, J.H. et. al.: Regional hyperthermia combined with radiotherapy for uterine cervical cancers: a multiinstitutional prospective randomized trial of the international atomic energy agency. Int. J. Rad. Oncol. Biol. Phys. 61, 145-153 (2005)

[vi]         van der Zee J, Gonzalez Gonzalez D, van Rhoon GC, van Dijk JD, van Putten WL, Hart AA. 2000. Comparison of radiotherapy alone with radiotherapy plus hyperthermia in locally advanced pelvic tumors: a prospective, randomised, multicentre trial. Dutch Deep Hyperthermia Group., The Lancet; 355(9210):1119–1125.

[vii].       Hasegawa T, Gu Y-H, Takahashi T, Hasegawa T, Yamamoto IY. (2003) Enhancement of Hyperthermic effects using rapid hyperthermia, In: Kosaka M, Sugahara T, Schmidt KL, Simon E. (eds.) Theoretical and experimental basis of Hyperthermia. Thermotherapy for Neoplasia, Inflammation, and Pain, Springer Verlag Tokyo, pp. 439–444

[viii].      Daniel, R.M.; Danson, M.J. Temperature and the catalytic activity of enzymes: A fresh understanding, FEBS Letters, 2013, 587(17): 2738-2743.
